# Prevention of Secondary Injury after Anterior Cruciate Ligament Reconstruction: Relationship between Pelvic-Drop and Dynamic Knee Valgus

**DOI:** 10.3390/ijerph20043063

**Published:** 2023-02-09

**Authors:** Rosario D’Onofrio, Anas Radi Alashram, Giuseppe Annino, Matteo Masucci, Cristian Romagnoli, Elvira Padua, Vincenzo Manzi

**Affiliations:** 1Member of the Medical-Scientific Multidisciplinary Commission, Italian Football Doctors Association-L.A.M.I.CA., 04023 Formia, Italy; 2Department of Physiotherapy, Faculty of Allied Medical Science, Middle East University, Amman 11622, Jordan; 3Centre of Space Bio-Medicine, Department of Medicine Systems, University of Rome “Tor Vergata”, 00133 Rome, Italy; 4Department of Kinesiology and Health Sciences, University of Waterloo, Waterloo, ON N2L 3G1, Canada; 5Sport Engineering Lab, Department Industrial Engineering, University of Rome “Tor Vergata”, 00133 Rome, Italy; 6Department of Human Sciences and Promotion of the Quality of Life, San Raffaele Roma Open University, 00166 Rome, Italy; 7Department of Humanities Science, Pegaso Open University, 80143 Naples, Italy; 8Hellas Verona Football Club, Via Olanda 11, 37135 Verona, Italy

**Keywords:** anterior cruciate ligament reconstruction, anterior cruciate ligament injuries, return to sport, knee joint, pelvis

## Abstract

(1) Background: Optimal neuromuscular, Lumbo-Pelvic-Hip Complex, and lower extremity control are associated with decreased risk factors for secondary anterior cruciate ligament (ACL) injury. This study aimed to analyze any asymmetries and malalignments in the Lumbo-Pelvic-Hip Complex and the lower limbs at 6 months after ACL reconstruction (ACLR). (2) Methods: We conducted an exploratory retrospective observational single-center study in patients during the outpatient postoperative rehabilitation program at ICOT (Latina, Italy). From January 2014 to June 2020, 181 patients were recruited, but only 100 patients (86 male 28 ± 0.6 years, 178 ± 0.5 height; 14 female 24 ± 2.0 years, 178 ± 3.0 height) were eligible for the inclusion criteria and studied 6 months after ACL reconstruction surgery. (3) Statistical analysis: Student’s *t*-tests and Pearson’s product-moment correlation coefficient were used to determine significant differences between affected and non-affected limbs and variables’ association. (4) Results: The study shows a decrease in neuromuscular control of the Lumbo-Pelvic-Hip Complex and dynamic adaptive valgus of the knee at 6 months after ACLR (mean difference between pathological and healthy limb of dynamic adaptive valgus was −10.11 ± 8.19° 95% CI −14.84 to −9.34; mean value was 16.3 ± 6.8° 95% CI 14.04 to 18.55 for healthy limb and 4.2 ± 3.1° 95% CI 3.15 to 5.21 for pathological limb, *p* < 0.0001). The results also showed a relationship between dynamic adaptive valgus and contralateral pelvic drop (r = 0.78, 95% CI 0.62 to 0.88, magnitude of correlation very large). (5) Conclusions: The analysis showed an associative correlation between decreased postural control of the pelvic girdle and dynamic adaptive valgus of the knee in 38% of patients; this study highlights the usefulness of the Single-Leg Squat Test (SLST) as a clinical/functional assessment to evaluate the rehabilitation process and as a preventive tool to reduce the risk of second ACL injuries during the return to sport.

## 1. Introduction

An anterior cruciate ligament (ACL) injury occurs due to trauma during athletic participation and frequently results in an inability to return to pre-injury activity levels [1]. The incidence of primary ACL injury is 1.7% per year in the athletic population [2]. Following an ACL injury, athletes mainly complain of knee joint instability [3], for which ACL reconstruction (ACLR) has become the recent gold standard of operative management [4]. It was concluded that athletes who can resume their sporting activity are more likely to be satisfied with the outcome of ACLR [5]. Recently, many surgical procedures, fixation methods, and rehabilitation protocols have been linked to ACLR [6]. Patients who underwent ACLR were approximately six times more likely to sustain an ACL injury within the first year after returning to their sports. Female athletes having undergone ACLR were almost five times more likely to sustain an ACL injury than female athletes with no history of an ACL injury [7]. Many risk factors for primary ACL injury have been identified. Female sex [8], race [9], and practicing pivoting sports [10] have been reported as risk factors for a primary ACL tear. Other risk factors have been identified, such as increased posterior tibial slope [11], narrow notch width [11], small size ACL [12], limb malalignments (i.e., Lumbo-Pelvic-Hip Complex asymmetries) [12], neuromuscular control [12], vertical directed and short femoral tunnel length [13], and graft tunnel length [14]. Return to sport after ACLR is an arduous decision-making process that needs to be structured based on shared and homogeneous scientific assessments [1]. Periodic short and long-term follow-ups to examine the impact of the injury on return to sport must be included in the Return to Performance [1]. Continuum rehabilitation protocols incorporate clinical assessment of the status of the healing process and functional testing; however, there is little predictive ability on the ultimate success [15]. Neuromuscular control deficits are common after ACLR. It has been associated with an increased risk of revision of ACLR and contralateral ACL injury [16,17]. Abnormal kinematics such as Lumbo-Pelvic-Hip Complex asymmetries and lower limb malalignment associated with injury are potentially identifiable using the Single Leg Squat test (SLST). Pelvic drop is associated with excessive knee valgus [18]. Additionally, patellofemoral pain syndrome (PFPS) is linked with excessive hip adduction, tibial external rotation, knee valgus, ankle pronation, pelvic obliquity, and ipsilateral trunk lean [18]. Females have a greater risk of anterior cruciate ligament injury than males because they exhibit excessive hip adduction, hip internal rotation, and knee valgus [18]. Dynamic functional tests (e.g., crossover hop for distance, triple hop for distance, single-leg squat) are reported in the literature as an assessment tool to validate return-to-sport decision-making [1]. The European Board of Sports Rehabilitation recommends hopping performance tests as a screening evaluation [19]. Hopping performance tests may express a difference of ≤10% between affected and non-affected limbs. Recently, a battery of functional tests has been established to facilitate decision-making regarding the return to sport after ACLR [19]. A study by Hewett et al. (2012) aimed to define modifiable risk factors for ACL injury, how these factors can best be modified, and when is the best time to diminish these risk factors [18]. It remains to be emphasized that the evaluation of neuromuscular control of the dynamic valgus of the knee and pelvic girdle must be part of periodic clinical and functional follow-ups [18]. Oleksy et al. (2018) suggest that an improved understanding of the potential contribution of lumbopelvic-hip problems in relation to a knee injury is needed for the development of more effective knee rehabilitation and injury prevention programs [20]. Hence, the role of Lumbo-Pelvic-Hip Complex asymmetries and lower limb malalignment in secondary injury following ACLR remains ambiguous. Therefore, the aim of this study was to investigate Lumbo-Pelvic-Hip Complex asymmetries and lower limb malalignment using the Single-Leg Squat Test (SLST) to understand their roles in secondary injury prevention after ACLR in athletes who return to high-level pivoting sports. Assessment of the pelvic girdle and especially the pubic region in athletes through functional/postural assessment tests such as the SLST are extremely important as they allow clinicians to identify the athlete at risk of injury and reduce the athlete’s predisposition to injury and post-injury complications.

## 2. Materials and Methods

### 2.1. Design 

We conducted an exploratory observational retrospective single-centered study. Our study is reported following the Strengthening the Reporting of Observational Studies in Epidemiology (STROBE) statement [21] (Appendix A).

### 2.2. Participants 

Operated people fulfilling inclusion criteria and participating in the outpatient post-operative rehabilitation program at ICOT (Latina, Italy) for a period of 5 years were consecutively included. Patients were included if: (a) aged from 20 to 40 years, (b) injured during sports activity, and (c) underwent ACLR using autografts semitendinosus gracilis free (STGF), semitendinosus gracilis with preserved insertion (STGPI) or bone-patellar tendon-bone (BPTB). Patients with a history of hip groin pain, sports hernia, muscle injuries, patellar tendinopathies, ankle joint injuries, and functional or structural dysmetria were excluded from this study. In fact, different studies showed that patients with Patello Femoral Pain Syndrome (PFPS) exhibit greater ipsilateral trunk tilt, contralateral pelvic drop, hip adduction, and knee abduction in the SLST than those without PFPS [22,23,24]. 

In addition, both muscle fatigue and decrements in the range of motion of dorsal ankle flexion are associated with greater trunk flexion, pelvic obliquity, pelvic tilt, pelvic rotation, and hip adduction [25,26]. 

To confirm functional dysmetria, we measured the distance from the umbilicus to the median malleolus and compared it with the contralateral leg. Moreover, to find out whether the lower limb dysmetria is structural, we measured the distance from the anterior superior iliac spine (SIAS) to the midpoint of the median (or internal) malleolus, comparing it to the measurement of the contralateral leg [27].

Demographic information: data on age, sex, and height were collected and analyzed by an experienced clinical assessor. All participants signed an informed consent form before they participated in the current study. The study was approved by the Internal Research Board of “Tor Vergata”, University of Rome. All the procedures involved in this study were in accordance with the Declaration of Helsinki.

### 2.3. Outcome Measures

#### 2.3.1. Q-Angle Assessment

Before the procedure, all patients underwent a Q-angle assessment to identify valgus or varus knee values. The intra-class coefficient (ICC) score ranged from 0.72 to 0.83 [28]. In the upright position with the feet in a neutral, anatomic landmark including the superior and inferior border of the patella, the tibial tubercle, anterior superior iliac spine (ASIS), and the center of the patella were identified through palpation and marked by a red adhesive marker. The Q angle was subsequently reported through analysis using Dartfish Motion Analysis software (Dartfish©, Fribourg, Switzerland). The procedure was recorded using a high-speed camera (Casio Exilim EX-ZR 3700—Japan) set at 120 Hz (time resolution ~8 ms) that was positioned at a distance of 2 m perpendicular to the frontal plan of the subject. The non-affected limb was evaluated first, followed by the affected limb. The evaluation was performed at the baseline (before surgery) and a 6-month follow-up after ACLR.

#### 2.3.2. Single-Leg Squat Test (SLST)

In the upright position with the feet in a neutral, patients were asked to place their hands on the iliac crests while remaining in a single support stance, first on the non-affected limb and then on the affected limb. The contralateral limb (non-supported) was in 45° hip flexion and 90° knee flexion. Subsequently, the patient was asked to perform the SLST [29].

The single-leg squat test (SLST) is commonly used for assessing overall biomechanical function. The SLST is a qualitative observational assessment [29]. The ICC score ranged from 0.75 to 0.90 [30]. The patient is asked to stand on one limb while flexing the opposite knee to 90° (Figure 1A,B). Subsequently, the patient is asked to place his hands on his hips while performing a single leg squat at 30° knee flexion, hold for 1–2 s, and then return to a fully extended knee position. Visual observation is used to assess whether patients have reached 30° of knee flexion. In case the patient does not reach 30° of knee flexion, the assessor gives verbal cues to increase or decrease the amount of knee flexion in the subsequent squats. The test was performed three times for each leg. 

The time was recorded using a stopwatch, and the joint angle was measured using a goniometer. The procedure was recorded using a digital camera that allows multiple-shot motion pictures. The final evaluation was performed using the Dartfish Motion Analysis Software. The video was analyzed by a clinician who also collected the data and estimated the functional postural parameters. The reliability, accuracy, and validity of the video analysis used in the SLST are documented, as preliminary scientific studies suggest [31,32]. 

Poor performance is identified by the presence of ipsilateral trunk tilt, contralateral pelvic drop, hip adduction, internal rotation, and knee valgus. The evaluation was performed at the baseline (before surgery) and a 6-month follow-up after ACLR. The SLST demonstrated moderate to excellent reliability for the evaluation of test performance [33,34,35].

### 2.4. Statistical Analysis

The results are expressed as mean ± SDs, and 95% confidence intervals (CI*s*) are presented where appropriate. The assumption of normality was verified using the Shapiro–Wilk W-test. Student’s *t*-tests (unpaired design) were used to determine significant differences between affected and non-affected limbs. Variables’ association was assessed using Pearson’s product-moment correlation coefficient (r) and provided with the corresponding confidence interval of 95%. The qualitative magnitude of associations was reported as follows: trivial r < 0.1, small r = 0.1–0.3, moderate r = 0.3–0.5, large r = 0.5–0.7, very large r = 0.7–0.9, nearly perfect r > 0.9, and perfect r = 1. The alpha level of significance was set at 0.05. All data were analyzed using SPSS Statistics version 22 for Windows (IBM Corp., Armonk, IL, USA).

## 3. Results

Overall, 100 participants (86% male), with a mean age of 27.44 years, exposed to either Patellar Bone-Tendon-Bone (PBTB) or semitendinosus–gracilis graft (STGT) ACLR approach completed the study. All subjects completed the study without experiencing injuries during the testing and rehabilitation phases. Demographic and health-related characteristics of the patients are presented in Table 1.

The evaluation screening showed a postural dysfunction in 38 patients (38%). Compared to the initial baseline values (non-affected limb, Q angle values ≤ 10 degrees, anterior superior iliac spine [ASIS] parallelism), these patients showed an associated decrease in postural control of the pelvic girdle (contralateral pelvic drop) and the knee (dynamic adaptive valgus) (Figure 2).

The mean total pelvic tilt referred to the 38 patients was 11.6 ± 5.7°. Although that figure, when referred to dynamic adaptive valgus, was 16.3 ± 6.8° (mean baseline Q-angle values, healthy limb in monopodial support of 4.6 ± 3.1°). It was noted that 12 of 38 patients showed an adaptive valgus greater than 20 degrees. Although 8 of 38 showed a pelvic tilt greater than 15 degrees. The correlative value of the dysfunctional peak was found only in one patient who had a pelvic tilt of 28 degrees and a dynamic adaptive valgus of 28 degrees. A peak of adaptive valgus was found during the test of 29 degrees (Q-angle baseline value of 6°). In addition, a contralateral pelvic drop peak was found during the 28-degree test (baseline value 180° ASIS parallelism-decrement value 152°). In addition, a positive correlation was found between decreased postural control of the pelvic girdle (contralateral pelvic drop) and dynamic adaptive valgus of the knee 6 months after ACLR (Figure 3).

## 4. Discussion

The main and novel finding of the study is that SLST can be considered a functional diagnostic test to indicate many dysfunctional movements within the kinetic chain, pelvic girdle, valgus/varus knee, and subtalar hyper pronation. In fact, a positive correlation was found between decreased postural control of the pelvic girdle (contralateral pelvic drop) as measured by SLST and dynamic adaptive valgus of the knee at 6 months after ACLR (Figure 3). Compared to the initial baseline value, approximately 40 percent of patients showed a postural dysfunction of the Lumbo-Pelvic-Hip Complex and lower limb malalignment at 6 months after ACLR (Figure 2). In this group, no significant percentage variation with respect to postural dysfunction was observed between athletes operated on with PBTB and with SGTF autograft (31/82 and 7/18 subjects, respectively). 

The literature showed that SLST defines the issues between kinetic chains across joint pathologies [36,37]. The unilateral nature of the test has encouraged; over time, researchers and practitioners have studied the evaluative applicability even in the post-ACLR. Moreover, the SLST remains a typical gestural scheme of the technical-athletic expressions of many sports [1,15]. Thus, it is commonly used as a tool for the evaluation screening of functional movements, in particular by the National Academy of Sports Medicine (NASM) [38]. The NASM values its use, in association with the Overhead Squat Test, to provide quality functional movement assessment related to sport-specific gestures. However, incorrect SLST application can lead to inaccurate results from the neuromuscular and biomechanical aspects, especially in the pelvic girdle [17,28,39]. 

Improvements in neuromuscular control of the valgus during SLST correlate with decreased pain and improvements in joint function [24]. Abductor weakness and hip joint mobility restrictions [40,41] increase the risk of ACL injury correlating with increased trunk postural/functional compensation, pelvic misalignment, increased hip adduction, and correlative knee valgus [22]. Therefore, we can state that the SLST can be used to identify neuromuscular risk factors to prevent secondary ACL injuries and return to sport [1,15]. 

In agreement with recent studies, there is a correlation between stability and neuromuscular control of the pelvic girdle and lower limb. It is confirmed that SLST can be a functional diagnostic assessment to indicate many dysfunctional movements within the kinetic chain, pelvic girdle, valgus/varus knee, and subtalar hyper pronation. Our study has shown that many athletes present with neuromuscular deficits at SLST 6 months after ACL reconstruction. Caution should, therefore, be used in allowing the athlete to engage in an unrestricted athletic activity [42]. Evidence supports that neuromuscular deficit at the SLST elevates risk factors for recurrence after ACL reconstruction surgery. Increased valgus loading of the knee associated with a deficit in pelvic girdle and trunk neuromuscular control is predictive of ACL injury [43]. 

Biomechanical changes resulting from lower limb misalignment can have a negative influence on joint load, mechanical efficiency of the tendon muscle apparatus and proprioceptive orientation, and consequently, on hip and knee feedback [44]. These adaptations result in altered neuromuscular function and control of the lower extremities. It is widely accepted that excessive frontal plane movement of the knee in the valgus contributes to an increased risk of anterior cruciate ligament injury [1,45,46].

The excessive frontal plane motion of the knee and associated peak valgus is linked with increased capsular ligamentous injuries of the knee [1,15,16]. Researchers have pointed out also that women often exhibit greater frontal plane knee motion during dynamic activities than men [1,15]. Differences in neuromuscular control of the hip have been cited as an important source of sex differences in lower extremity movement patterns [16,39]. In this regard, the comparative risk of injury from exposure is 2 to 8 times greater in women than in men [39]. Dynamic knee stability is achieved through neuromuscular control of a multifactorial kinetic chain because the knee is directly supported by the surrounding muscles. Additionally, the knee depends on the more proximal muscles of the hip and trunk [17].

Thus, we can say that it is important to evaluate the rehabilitation process through periodic follow-ups, based on the scientific evidence present in the current literature [1,43,47], to return to pivoting and cutting sports. It remains important to consider the risk factors of complications earlier and in future injuries, to study dynamic, synergistic pathological movements [1,43,47] and decrease the risk factors of future both revision ACLR and contralateral ACL injury. 

The postural dysfunction of excessive hip adduction, valgus knee, pelvic obliquity, and ipsilateral trunk tilt has often been associated with lumbar stress injuries [48], PFPS [38,40], and ACL injuries [23]. 

During technical-athletic gestures, the knee is often subject to excessive frontal plane motion due to hip internal rotation, femur adduction, and tibia external rotation [1,37].

The identification of the functional integrity of the entire pelvic girdle guarantees, in athletes, optimal dynamics of the spine and lower limb. The postural and functional evaluation tests are important because they allow intercepting the athlete at risk of injury. However, pelvic girdle assessment in athletes must be reliable and reproducible.

Didactically, once the right structural symmetries are re-established, it is possible to restore muscular balance and quality of movement [15], especially in the late stage of sports rehabilitation [15]. This is performed with a protocol based on evaluation, diagnostic tests, and evidence-based rehabilitation treatment choices [1,15]. 

We examined the Lumbo-Pelvic-Hip Complex asymmetries and lower limbs malalignment at 6 months after ACLR to understand their roles in secondary injury after ACLR and, consequently, help to prevent secondary injuries in an athlete who returns to high-level pivoting sports. 

## 5. Limitations of the Study

This study was limited by involving multiple clinicians and therapists in the rehabilitation process. This may lead to an exclusively observational study that did not allow us to define the causative factors due to a lack of homogeneity in the rehabilitation process and the consequent return to sport. Another limitation of this study relates to the use of a convenience sample (i.e., semi-professional team sport players) and not one which allows this investigation to produce further generalizations. Finally, despite the validity of the video-based evaluation, an integration with other investigation systems (e.g., radiological imaging) to assess the Lumbo-Pelvic-Hip Complex asymmetries and lower limb malalignment should allow the improvement of the accuracy of these measures.

## 6. Conclusions

This observational study has allowed us to study in a simple, practical, reliable, and reproducible way the posture-functional picture of the pelvic girdle and lower limb at 6 months after ACLR. The analysis showed an associative correlation between decreased postural control of the pelvic girdle and dynamic adaptive valgus. The results of this study highlight the usefulness of the SLST as a clinical/functional assessment to evaluate the rehabilitation process and as a preventive tool to reduce the risk of second ACL injuries during the return to sport.

This finding could deeply influence the choice of rehabilitation process planning used for patients after ACL reconstruction surgery.

## 7. Patents

This section is not mandatory but may be added if there are patents resulting from the work reported in this manuscript.

## Figures and Tables

**Figure 1 ijerph-20-03063-f001:**
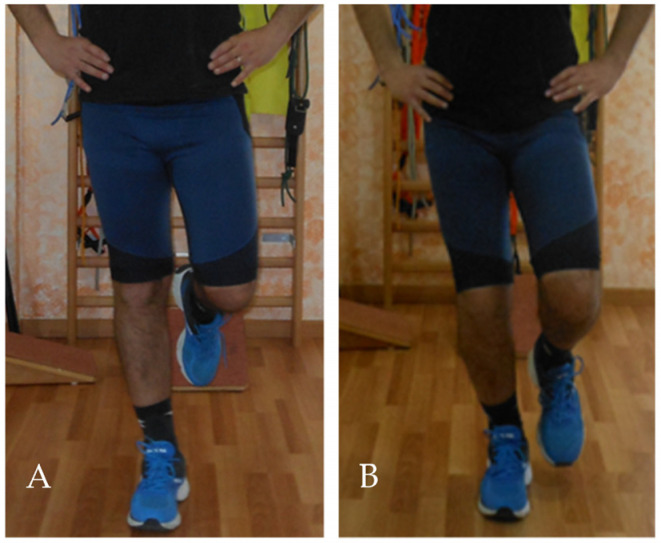
Single leg squat test, initial test position (**A**), and final test position (**B**).

**Figure 2 ijerph-20-03063-f002:**
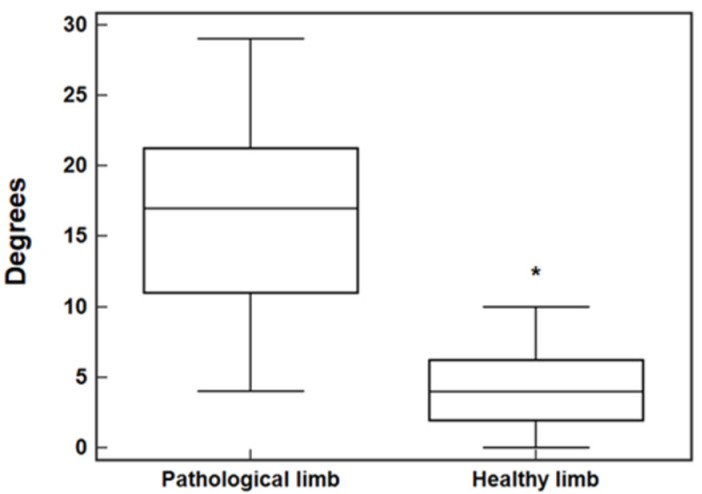
Difference between healthy and pathological limb. Values, expressed in degrees, of dynamic adaptive valgus (16.3 ± 6.8° 95% CI 14.04 to 18.55; 4.2 ± 3.1° 95% CI 3.15 to 5.21, *n* = 38, * *p* < 0.0001).

**Figure 3 ijerph-20-03063-f003:**
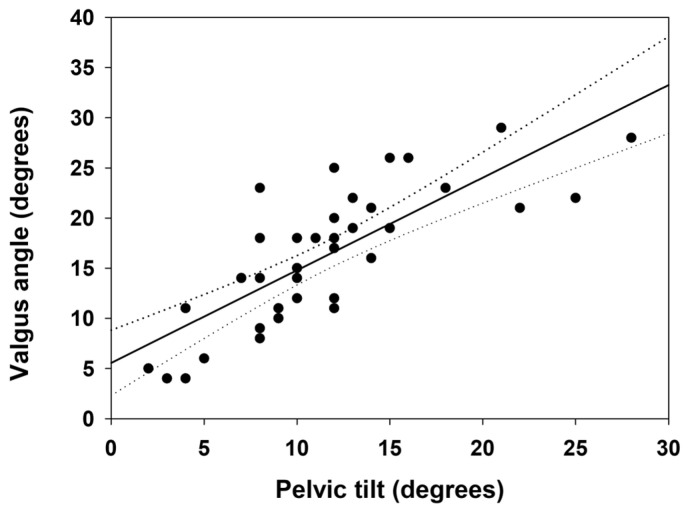
Correlation between valgus angle and pelvic tilt, in the frontal plane, during the Single Leg Squat Test (*n* = 38, r = 0.78, 95% CI 0.62 to 0.88).

**Table 1 ijerph-20-03063-t001:** Patient’s Characteristics. PBTB: Patellar Bone-Tendon-Bone; STGT: semitendinosus–gracilis graft.

Patients Characteristics	
Gender (*n*)	
Male	86
Female	14
Age (Mean ± SD)	
Male	28 ± 0.6
Female	24 ± 2.0
Height (Mean ± SD)	
Male	178 ± 2.5 cm
Female	172 ± 3.0 cm
ACL surgery approach (*n*)	
PBTB	82
STGT	18

## Data Availability

All study data are included in the present manuscript.

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
