# Peer review of "Prevention of Secondary Injury after Anterior Cruciate Ligament Reconstruction: Relationship between Pelvic-Drop and Dynamic Knee Valgus"

_ijerph, 2023, doi:10.3390/ijerph20043063_

Round 1

Reviewer 1 Report

Thank you for giving me this opportunity to review this article. The article is well written, though I have some serious concerns regarding the article.

Keywords: use MeSH keywords

Abstract:

  1. Mention the study design.
  2. The study duration is limited to 2018 – hence the reports submitted are outdated one.
  3. Mention the study setting.
  4. Include the character of the study participants.
  5. Include the outcome measures measured for the review.
  6. Mention the statistical tests performed for the study.
  7. The results should be presented with 95%CI (upper limit – lower limit) for all the variables.
  8. The objective and the conclusion of the study provides contradictory statements
  9. Conclusion should be drawn on the basis of the study reports, not on an assumption.

Manuscript

  1. How come this study is differing from the reference study 18?
  2. The novelty of the study is missing, including more recent references emphasizing the need for this study.
  3. Mention the gaps monitored by the researcher.
  4. Include the clinical significance of this study over clinicians, patients, and researchers after the study hypothesis.
  5. The study duration is limited to 2018 – hence the reports submitted are outdated one.
  6. Include the character of the study participants in detail.
  7. Is SLST is intervention or outcome measure?
  8. Include the ethical and review registration number.
  9. Mention who has included the study participants in the trial?
  10. Include the manufacturer details of the software used Dartfish motion analysis.
  11. Include the study outcome measures and its reliability and validity. 
  12. Mention clearly what measurements were taken for lumbar, pelvis, hip and lower limb malalignments.
  13. Include the reference study for sample size calculation.
  14. The samples included in the study is not sufficient enough to generalize the reports.
  15. The study includes only the analysis of pelvic angle and knee valgus deformity – what about the deformities in the lumbar, pelvis and hip region?  
  16. There is redundant and unnecessary information in the discussion part and not presented in a logical manner. 
  17. Add more recent references explaining the mechanism of changing outcome variables after ACLR surgery.
  18. The conclusion should be more concise and self-explanatory and drawn on the basis of study reports. 
  19. Add more real-time limitations faced by the researcher and the study. 
  20. Include future recommendations of the study.

Author Response

Thank you for giving me this opportunity to review this article. The article is well written, though I have some serious concerns regarding the article.

Keywords: use MeSH keywords

Changed to MeSH terms.

Abstract:

  • Mention the study design.

It was mentioned.

  • The study duration is limited to 2018 – hence the reports submitted are outdated one.

It was corrected

  • Mention the study setting.

It was mentioned.

  • Include the character of the study participants.

It was included.

  • Include the outcome measures measured for the review.

It was included.

  • Mention the statistical tests performed for the study.

It was added.

  • The results should be presented with 95%CI (upper limit – lower limit) for all the variables.

It was mentioned

  • The objective and the conclusion of the study provides contradictory statements

The objective was modified.

  • Conclusion should be drawn on the basis of the study reports, not on an assumption.

The conclusion was modified.

 Manuscript 

  • How come this study is differing from the reference study 18?

Clarified at the end of the introduction.

  • The novelty of the study is missing, including more recent references emphasizing the need for this study.

Clarified at the end of the introduction. We did not find additional recent references like our subject.

  • Mention the gaps monitored by the researcher.

Clarified at the end of the introduction.

  • Include the clinical significance of this study over clinicians, patients, and researchers after the study hypothesis.

done

  • The study duration is limited to 2018 – hence the reports submitted are outdated one.

We agree with your observation but due to some important events not attributable to our will, we concluded this study only recently. In any case, considering that the elapsed time cannot influence the results of this study we thought it appropriate to omit the reference period in the manuscript if you agree   

  • Include the character of the study participants in detail.

Clarified in the results section.

  • Is SLST is intervention or outcome measure?

It was used as an outcome measure only and we modified the text accordingly as reported in the methods.

  • Include the ethical and review registration number.

Considering the non-invasive nature of the study, the ethics committee considered appropriate only the consent of the Internal Research Board of the "Tor Vergata" University of Rome which does not have a registration number. This procedure has already been accepted for other articles already published by our research group in the MDPI journals. I remain at your disposal for further clarification.

  • Mention who has included the study participants in the trial?

Mentioned at beginning of the results section.

  • Include the manufacturer details of the software used Dartfish motion analysis.

Done

  • Include the study outcome measures and its reliability and validity.

Done

  • Mention clearly what measurements were taken for lumbar, pelvis, hip and lower limb malalignments.

Done

  • Include the reference study for sample size calculation.

We did not estimate the sample a priori with a specific method because, being our observational study on ACLR athletes, the analyzed data were collected from year to year. All this will be included within the limits of the study.

  • The samples included in the study is not sufficient enough to generalize the reports.

According to your observation, we include this limitation in the body of the manuscript (limitation section).

  • The study includes only the analysis of pelvic angle and knee valgus deformity – what about the deformities in the lumbar, pelvis and hip region? 

The request of the reviewer is appropriate but in our study, we did not investigate other deformities except those reported in the manuscript. 

  • There is redundant and unnecessary information in the discussion part and not presented in a logical manner.

According to your suggestion, we modified the discussion part

  • Add more recent references explaining the mechanism of changing outcome variables after ACLR surgery.

Added

  • The conclusion should be more concise and self-explanatory and drawn on the basis of study reports.

We modified the text accordingly

  • Add more real-time limitations faced by the researcher and the study.

Added

  • Include future recommendations of the study.

Added

Reviewer 2 Report

Thank you for the opportunity to review this manuscript. The authors work on a highly-sought-after topic of an ideal return-to-sport after ACLR, which would be of high interest to the relevant community. To do this, the authors analyzed "any asymmetries and malalignments in the lumbo-pelvic hip Complex and the lower limbs at 6 months after ACL reconstruction". Unfortunately, one cannot see from the abstract how this extensive investigation was carried out. In the further text, a video-based evaluation is mentioned with a software solution assessment, whether this can fully do justice to the broad question of the study would have to be discussed more extensively and should be clearly stated in the limitations section. It would also be interesting to know which radiological imaging methods the examiners had access to or if these were included in their findings.

The number of evaluated patients also remains questionable in the abstract. "181 patients were analyzed and studied 6 months after ACL reconstruction surgery. Of them, 100 patients (86% men) met the eligible criteria." If 181 have already been evaluated, what is the composition of the number of exclusions? Due to the high frequency of muscle injuries and ankle injuries, this reason for exclusion is also surprising. How were these exclusions diagnosed, how recent did they have to be?
What does it mean that the clinician "estimated the performance parameters"?
Was the mentioned age range 20-40 randomly chosen? Is there information on other relevant issues such as post-treatment scheme, re-rupture rate, PROMs?

The choice of one functional test is also surprising. Recent literature might suggest jump tests as seemingly useful in this context (E.g. Kotsifaki A, Van Rossom S, Whiteley R, Korakakis V, Bahr R, Sideris V, Jonkers I. Single leg vertical jump performance identifies knee function deficits at return to sport after ACL reconstruction in male athletes. Br J Sports Med . 2022 May;56(9):490-498. doi: 10.1136/bjsports-2021-104692. Epub 2022 Feb 8. PMID: 35135826; PMCID: PMC9016240.).

A detailed evaluation regarding the graft used would also be useful. Even if BTB was mostly used in this collective, it would also be relevant here to see differences to the smaller group with STGT graft. Other studies of a similar nature show relevant postoperative differences between using tendons of the extensor apparatus or hamstrings grafts of the flexor apparatus. Of course, a longer follow-up period would also be advantageous here.

Furthermore, some structural issues arise: Different formatting styles in the title, multiple different spellings e.g. of "lumbo-pelvic hip complex" and also frequent changes between singular and plural, e.g. in lines 64-70. Small numbers should be written out throughout the manuscript.

All in all, this study manuscript presents a quite interesting question, and also some interesting and relevant results, even if the manuscript still needs thorough revision and definite limitations should be stated more clearly.

Author Response

Thank you for the opportunity to review this manuscript. The authors work on a highly-sought-after topic of an ideal return-to-sport after ACLR, which would be of high interest to the relevant community. To do this, the authors analyzed "any asymmetries and malalignments in the lumbo-pelvic hip Complex and the lower limbs at 6 months after ACL reconstruction".

  • Unfortunately, one cannot see from the abstract how this extensive investigation was carried out.

The abstract was modified accordingly

  • In the further text, a video-based evaluation is mentioned with a software solution assessment, whether this can fully do justice to the broad question of the study would have to be discussed more extensively and should be clearly stated in the limitations section.

Following your suggestion we add the issue in the limitation section

  • It would also be interesting to know which radiological imaging methods the examiners had access to or if these were included in their findings.

No radiological imaging methods were included in this study

  • The number of evaluated patients also remains questionable in the abstract. "181 patients were analyzed and studied 6 months after ACL reconstruction surgery. Of them, 100 patients (86% men) met the eligible criteria." If 181 have already been evaluated, what is the composition of the number of exclusions?

According to the reviewer's suggestion, we corrected the composition of the number of patients evaluated in the abstract.

  • Due to the high frequency of muscle injuries and ankle injuries, this reason for exclusion is also surprising. How were these exclusions diagnosed, how recent did they have to be?

 All patients with a prior history of ankle or muscle injury were excluded from the study to avoid invalidating the SLST, regardless of the time of injury

  • What does it mean that the clinician "estimated the performance parameters"? Was the mentioned age range 20-40 randomly chosen? Is there information on other relevant issues such as post-treatment scheme, re-rupture rate, PROMs?

The performance parameters are related to spatial position of the markers analysed in the study and mentioned in the text (line 176….). Considering the conventional type of sample we included typically the agonist sports athletes from low to high level. Related to your last question, the observational nature of the study allow us to involve multiple clinicians and therapists without define the causative factors due to a lack of homogeneity in the rehabilitation process and the consequent return to sport  

  • The choice of one functional test is also surprising. Recent literature might suggest jump tests as seemingly useful in this context (E.g. Kotsifaki A, Van Rossom S, Whiteley R, Korakakis V, Bahr R, Sideris V, Jonkers I. Single leg vertical jump performance identifies knee function deficits at return to sport after ACL reconstruction in male athletes. Br J Sports Med. 2022 May;56(9):490-498. doi: 10.1136/bjsports-2021-104692. Epub 2022 Feb 8. PMID: 35135826; PMCID: PMC9016240.).

We thank the reviewer for the suggestion but we promote this study before and six months after the surgery and we considered the SLST more suitable and safe overall in the preceding surgery intervention phase

  • A detailed evaluation regarding the graft used would also be useful. Even if BTB was mostly used in this collective, it would also be relevant here to see differences to the smaller group with STGT graft. Other studies of a similar nature show relevant postoperative differences between using tendons of the extensor apparatus or hamstrings grafts of the flexor apparatus. Of course, a longer follow-up period would also be advantageous here.

We thank the Reviewer for the appropriate question. We add this information in the text accordingly (in the discussion section)       

  • Furthermore, some structural issues arise: Different formatting styles in the title, multiple different spellings e.g. of "lumbo-pelvic hip complex" and also frequent changes between singular and plural, e.g. in lines 64-70. Small numbers should be written out throughout the manuscript.

Done

  • All in all, this study manuscript presents a quite interesting question, and also some interesting and relevant results, even if the manuscript still needs thorough revision and definite limitations should be stated more clearly.

 According to the reviewer’s suggestion, the study limitations has been included in the body of the manuscript.

Round 2

Reviewer 2 Report

The authors have put some effort into improving the manuscript according to the review. Nevertheless, some very important weaknesses remain. The evaluation of the ambitious study objective "to analyze any asymmetries and malalignments in the Lumbo-Pelvic-Hip Complex and the lower limbs at 6 months after ACL reconstruction (ACLR)" is limited - now clearly stated by the authors - to the video recordings with a single camera. Here, a derivation should be given in the introduction or in the methodology part, why the authors are of the opinion that this methodology is adequate for the above-mentioned question and whether preliminary scientific work suggests any reliability of this method.

Even if a few other (manual?) measurements are mentioned later in the manuscript radiological measurements would be the usual standard here. The statement that the clinician "estimated the test performance parameters" (line 186f) is still noted in the manuscript and remains highly questionable in terms of reliability.

Additional information on the graft choice was added and the selection of patients was also presented slightly better, even if not all of the questions from the review were answered. With a few exceptions, the main terms are now spelled in the same way throughout the manuscript.

Other questions of the review were not (clearly) answered: Was the mentioned age range 20-40 randomly chosen? Re-rupture rate? 

The exclusion criteria of a prior ankle or muscle injury (no matter how old) is surprising, as previously noted, since both injuries are among the most common injuries of the musculo-skeletal system and subsequent problems rarely persist in a sporting collective. In particular, this raises the question of a recall bias in the group, which did not provide any information in this regard.

Please correct lines 124-126: "In fact, different studies have shown showed that patients with Patello Femoral Pain Syndrome (PFPS) exhibit greater ipsilateral trunk tilt, contralateral pelvic drop, hip adduction, and knee abduction in the SLST than those without PFPS [22,23,24]."

Author Response

The authors have put some effort into improving the manuscript according to the review. Nevertheless, some very important weaknesses remain. The evaluation of the ambitious study objective "to analyze any asymmetries and malalignments in the Lumbo-Pelvic-Hip Complex and the lower limbs at 6 months after ACL reconstruction (ACLR)" is limited - now clearly stated by the authors - to the video recordings with a single camera. Here, a derivation should be given in the introduction or in the methodology part, why the authors are of the opinion that this methodology is adequate for the above-mentioned question and whether preliminary scientific work suggests any reliability of this method.

Thanks to your suggestions we hope to have improved the soundness of the manuscript

Even if a few other (manual?) measurements are mentioned later in the manuscript radiological measurements would be the usual standard here. The statement that the clinician "estimated the test performance parameters" (line 186f) is still noted in the manuscript and remains highly questionable in terms of reliability.

AR:

Following to your suggestion we modified the text

Additional information on the graft choice was added and the selection of patients was also presented slightly better, even if not all of the questions from the review were answered. With a few exceptions, the main terms are now spelled in the same way throughout the manuscript.

Other questions of the review were not (clearly) answered: Was the mentioned age range 20-40 randomly chosen? Re-rupture rate? 

AR:

We considered this age range which included still physically and competitively active subjects with a similar biological profile in order to also ensure an adequate number of subjects for the study. In addition, we add in the results section that none of the subjects included in the study reported injuries during the rehabilitation phase

The exclusion criteria of a prior ankle or muscle injury (no matter how old) is surprising, as previously noted, since both injuries are among the most common injuries of the musculo-skeletal system and subsequent problems rarely persist in a sporting collective. In particular, this raises the question of a recall bias in the group, which did not provide any information in this regard.

AR:

The sample selected for the study remained the same at all stages of the trial. The exclusion criteria were only adopted in the selective phase of the sample before the start of the trial. None of the subjects included in the study reported injuries during the rehabilitation phase. This detail has been inserted in the text.

Please correct lines 124-126: "In fact, different studies have shown showed that patients with Patello Femoral Pain Syndrome (PFPS) exhibit greater ipsilateral trunk tilt, contralateral pelvic drop, hip adduction, and knee abduction in the SLST than those without PFPS [22,23,24]."

done
